# The Experiences and Perspectives of Persons with Prostate Cancer and Their Partners: A Qualitative Evidence Synthesis Using Meta-Ethnography

**DOI:** 10.3390/healthcare12151490

**Published:** 2024-07-27

**Authors:** Seidu Mumuni, Claire O’Donnell, Owen Doody

**Affiliations:** 1Department of Nursing and Midwifery, University of Limerick, V94 T9PX Limerick, Ireland; seidu.mumuni@ul.ie (S.M.); claire.odonnell@ul.ie (C.O.); 2Health Research Institute, University of Limerick, V94 T9PX Limerick, Ireland

**Keywords:** prostate cancer, partners, experiences, qualitative evidence synthesis (QES)

## Abstract

Prostate cancer affects one in nine men, so understanding patients’ and their partners experiences is crucial for developing effective treatments. The purpose of this review was to synthesis and report the experiences and views of persons with prostate cancer and their partners. Methods: A qualitative evidence synthesis (QES) was conducted following the eMERGe reporting guideline. Six databases were searched for the relevant literature, and the Critical Appraisal Skills Program (CASP) tool was used for quality appraisal. Results: A total of 1372 papers were identified, and 36 met the inclusion criteria. Four themes emerged: quality of life, relationships and dynamics, treatment journey and survivorship and aftercare. Conclusions: Prostate cancer’s impact on patients and partners is significant, requiring comprehensive support, holistic care, tailored assistance, and research into therapies to minimize adverse effects and address emotional distress and relationship strain. Prostate cancer treatment causes physical changes, triggering feelings of loss and grief, and affects coping mechanisms. Drawing on emotional support and education is vital for boosting confidence and resilience, as many patients and partners face fears of recurrence and lifestyle changes, highlighting the need for tailored information and presurgery support.

## 1. Introduction

Prostate cancer is the second most frequently occurring malignancy in men globally, responsible for around 7% of all cancer-related fatalities in men [1,2]. One in nine men will have a prostate cancer diagnosis at some point in their lives [3], and both the person with prostate cancer and their partners may be significantly impacted by a prostate cancer diagnosis [4]. The treatment of prostate cancer frequently entails invasive actions such as surgery, radiation therapy, or chemotherapy, which may have adverse effects that include erectile dysfunction and urine incontinence [5,6]. The experiences of prostate cancer patients and their partners highlight that both patients’ and their partners’ psychological health may be significantly impacted by receiving a prostate cancer diagnosis [7]. Moreover, the quality of life for both prostate cancer patients and their partners might be impacted by their shared experiences [8]. The impact of various therapeutic methods such as regular exercise and counselling (diet, spiritual, psychological, etc.) on the physical and mental health of prostate cancer patients and their partners has also been highlighted [1,9]. The significance of offering prostate cancer patients and their partners appropriate support throughout therapy/treatment and the requirement for additional assistance have been emphasized [10,11]. Prostate cancer patients and their partners experiences are nuanced and varied, and it is crucial to comprehend these experiences to deliver treatments that support prostate cancer patients and their partners throughout therapy/treatment.

Physical and cognitive functions play a critical role in determining quality of life among older adults with prostate cancer [2]. While physical limitations after treatment can significantly affect daily life and well-being, better cognitive function serves as a protective factor. This suggests that interventions targeting cognitive health alongside physical rehabilitation could potentially mitigate the negative impacts of reduced physical function [2]. There are profound psychological and emotional challenges faced by individuals with prostate cancer and their partners [3,6,7]. Gay and bisexual men experience heightened distress due to treatment-related side effects and societal stigma surrounding sexuality [3]. Pervasive anxiety, depression, and fears of cancer recurrence are evident among patients, impacting their self-esteem, social interactions, and overall quality of life [6]. Effective communication and enhanced emotional support emerge as critical for addressing these challenges adequately [7]. Both patients and partners struggle with discussing sexual issues openly, leading to feelings of isolation and diminished relationship satisfaction [3,4]. There can be a profound psychological impact of sexual dysfunction, including feelings of inadequacy and loss of identity for patients, and emotional neglect for partners. The indispensable role of partners and caregivers in the prostate cancer journey cannot be underestimated. Partners often assume caregiving responsibilities, navigating emotional stress and practical challenges [8,9]. They play crucial roles in supporting physical activity, advocating for comprehensive care, and managing the psychological well-being of patients [9,10]. Understanding and addressing the needs of partners, including their own emotional and physical health, is a critical component of holistic care approaches. The need for person-centered care models is highlighted to ensure that the diverse needs of patients with prostate cancer and their partners are adequately met throughout the treatment and recovery process [6,7,9,10].

Prostate cancer significantly affects both patients and their partners, encompassing a broad spectrum of emotional, physical, and relational dimensions. By systematically analyzing existing qualitative studies, this synthesis aims to uncover the impacts of prostate cancer and its treatments, shedding light on the physical and psychological burdens borne by patients and their partners. This approach acknowledges the critical role of partners in providing support, the challenges of navigating intimacy and communication after treatment, and the long-term emotional and relational complexities that arise. The goal of this qualitative evidence synthesis was to explore these multifaceted experiences and perspectives, enhance our understanding of these intertwined experiences, and inform the development of more effective, compassionate care strategies tailored to the unique needs of both patients and their partners. To the best of our knowledge, there has been no review published on the experiences of both persons with prostate cancer and their partners.

## 2. Materials and Methods

### 2.1. Study Aim

Our aim was to provide a comprehensive synthesis of qualitative literature reporting the experiences and perspectives of patients with prostate cancer and their partners.

### 2.2. Study Design

A qualitative evidence synthesis (QES) using meta-ethnographic methods was applied to represent the experience of patients with prostate cancer and their partners [12]. The review is reported in line with the eMERGe meta-ethnography reporting guideline [13].

### 2.3. Reflexivity

All stages of the review were conducted by the lead author (SM) under the guidance of the research team (OD, COD). The lead author is a registered nurse and Ph.D. candidate who had 9 years of clinical experience in hospital settings in Ghana and Ireland. The lead author was an advanced beginner as a qualitative researcher at the outset of the synthesis and had completed an integrative review and published a scoping review as part of their Ph.D. In addition, the lead author had completed postgraduate training in qualitative research and study days on meta-ethnography/qualitative evidence synthesis. Members of the wider research team had clinical experience as a general nurse (COD) and Intellectual disability nurse (OD). COD and OD were currently educationalists with experience in conducting primary qualitative research and qualitative data analysis and synthesis.

### 2.4. Search Selection

A systematic literature search of six databases was completed in January 2024 (Scopus, Academic Search Complete, MEDLINE, PsycINFO, Cumulative Index to Nursing and Allied Health Literature (CINAHL), Cochrane Library/Cochrane trial). A search strategy was developed based on three key concepts: ‘prostate cancer’, ‘person/partners’, and ‘experiences’; the search was not limited to qualitative research at the search stage, so all the literature could be screened. Databases were searched from the year 2000 to 2023, and the search process and outputs are available on the open platform figshare [14]. The search was guided by the inclusion and exclusion criteria outlined below and a search strategy (Table 1). The keywords were employed in (AB) abstract or author-supplied abstract searches, (TI) title searches, and using Boolean operators “AND” and “OR”. Each phrase had a “wildcard” denoted by an asterisk (*), which was used to expand the search and find pertinent data using alternatives to/variations in words.

### 2.5. Inclusion and Exclusion Criteria

Articles meeting all the following criteria were included:Qualitative design reporting recognized methods of qualitative data collection and analysis.Participants were either persons with prostate cancer and/or their partners.Study findings report perspectives and experiences of persons with prostate cancer and/or their partners.Language confined to English.

The following criteria were used for exclusion:Studies that used anything rather than qualitative data collection and analysis.Participants in Non qualitative studies.Studies that focused on other illnesses and cancers.Studies published in languages other than English.

### 2.6. Selection Procedure

Database search results were exported to Endnote Library 2022 (Clarivate Analytics, Pennsylvania, PA, USA) to identify and remove duplicates, and then transferred to Rayyan (Qatar Computing Research Institute) for screening and voting. Title and abstracts were screened independently against the inclusion criteria by two reviewers (SM, OD). Where necessary, disagreements were resolved through discussion and involvement of the third member of the team (COD). Two reviewers subsequently independently screened the full-text articles (SM, OD) for final decisions regarding inclusion with disagreements resolved through discussion and involvement of the third reviewer where necessary (COD). The reference lists of all included studies (backward chaining) were then reviewed to identify potential further studies for inclusion.

### 2.7. Critical Appraisal and Data Extraction

The methodological quality of the included studies was appraised using the ten-item Critical Appraisal Skills Programme (CASP) checklist for qualitative research [15]. Two evaluators (SM and OD) independently assessed the quality of each study against this checklist. Disagreements were managed through discussion or involvement of a third member of the research team (COD). The CASP appraisal reported high study quality with quality scores ranging from 8 to 10, and the full record is available on the open platform figshare [14].

No studies were excluded based on quality, as the appraisal was used to highlight methodological limitations in the interpretation of study findings. To pilot the data extraction form, two members (SM, OD) of the team extracted data from five of the included studies independently and met to discuss the use of the tool and to consider the consistency of extraction. The remaining data were extracted independently using the data extraction form, and, where clarification or discussion was needed, a meeting was held by the review team. The data extraction form included author, year, title, country, aim, methods (design, sample, data collection, data analysis), stage of illness, experience, key findings, and limitations.

### 2.8. Data Synthesis

A seven-step meta-ethnographic approach [12] was used to synthesize the findings across all studies, as it allowed the synthesis of findings from multiple studies and the development of a higher-order interpretation [16,17]. The first phase, ‘getting started’, involved the development of the review question and the setting of the inclusion/exclusion criteria by the research team. The second phase focused on ‘deciding what is relevant to the initial interest’. An agreement on the inclusion criteria was reached among the research team members, and searching for the research articles was conducted, followed which a quality appraisal was conducted on each included study. The third phase, ‘reading the articles’, involved reading and rereading all included studies and extracting second-order constructs (interpretation of an experience). Phase four involved ‘determining how the studies are related’, where the key concepts from each study (phrases or themes) were then contrasted against one another by the research team, allowing common and recurring concepts to be identified. Phase five involved ‘translating the studies into one another’, where the research team took the identified concepts and compared each concept with all other concepts to identify similarities and differences across concepts and to organize the concepts into further conceptual categories (presented in the Results Section as themes). This process was similar to the process of constant comparison and allowed the research team to stay close to the meaning and contexts of the original studies. Phase six of the analysis, ‘synthesizing translations’, involved a line of argument synthesized from the conceptual categories identified in phase five. Finally, phase seven, ‘expressing the synthesis’, was conducted according to the eMerge reporting guidance, which is provided in Appendix A [13], and the PRISMA flow diagram [18].

### 2.9. Public and Patient Involvement

No formal public and patient involvement (PPI) stakeholder panel of persons with prostate cancer or their partners were involved in the development of this study. However, three persons with prostate cancer and their partners contributed to the interpretation of the findings. When the preliminary analysis was complete, it was shared with the three people with prostate cancer and their partners who commented on the interpretation and presentation of findings to ensure they were in keeping with their experiences.

## 3. Results

A total of 1372 papers were found through database searches; after duplicates were removed (*n* = 701), 671 papers remained for screening. At the title and abstract screening, 509 papers were eliminated, and the remaining 81 papers went through to full-text review. The PRISMA flow diagram (Figure 1) describes the screening process, reasons for exclusion (*n* = 45), and identifies 36 papers included in this review. Searching of the grey literature was conducted but yielded no results that met the criteria for inclusion. The eMERGe meta-ethnography reporting guideline was used during the review.

### 3.1. Characteristics of Included Studies

The qualitative designs employed included the focus group methodology [19,20,21], phenomenological-hermeneutic approach [22,23,24], exploration study [25,26,27,28,29,30,31,32,33,34,35,36], cross-sectional qualitative study [37,38,39], phenomenological qualitative study [40,41], longitudinal study [36,42,43], exploratory descriptive study [44,45,46,47], qualitative content analysis [48], ground theory [49], observational study [50], and Giorgi’s descriptive phenomenological [51]. Two studies [52,53] failed to report the specific qualitative design used. Sample size ranged between 5 and 42 participants, and regarding data collection, individual interviews, open-ended surveys, and focus group interviews were used. Regarding countries of origin, nine were from the United States of America [19,21,29,36,38,45,49,52,53], six from Australia [25,28,31,34,37,54], five each from Canada [28,32,33,43,46] and the United Kingdom [26,27,35,41,50], three from Denmark [20,22,40], two each from Switzerland [39,48], Norway [23,24] and Sweden [30,47], and one from each of Ireland [51] and Brazil [44]. In terms of data analysis, qualitative content analysis [30,48], narrative analysis (interpretive phenomenological approach (Heidegger’s philosophy) [37,52], grounded theory methods [19,27,53], thematic analysis [21,25,26,28,29,31,32,34,35,36,38,44,45,47,49,50], Braun and Clarke’s 2006 method [33,46], a qualitative media analyzer [22], Giorgi’s phenomenological method [20,40,51], phenomenological hermeneutic framework [23,24], comparative method of grounded theory [39], non-numerical unstructured data indexing, searching, and theorizing [42,43] were used. All papers identified obtained ethical approval before conducting research.

### 3.2. Quality Appraisal

The results of the quality appraisal are available through an open repository [14]. No studies were excluded based on quality as all included studies contributed important information addressing the research question. Regarding the range of quality of the papers, the retrieved literature showed strong methodologies, rigorous designs, clear objectives, robust data collection and analysis methods, minimized biases, and transparent reporting. To ascertain the quality of papers, we looked at the study design, data collection, sample size and sampling, data analysis, reflexivity, and relevance of the paper. The CASP tool for qualitative studies was used to apprise the trustworthiness of the paper.

### 3.3. Themes

To develop appropriate themes for this study, we organized the findings into codes, then into subthemes, and finally into the appropriate themes (Table 2). The four themes that emerged were quality of life, relationships and dynamics, treatment journey and survivorship and aftercare.

#### 3.3.1. Quality of Life

The impact of prostate cancer treatments on a patient’s daily life is often overlooked by healthcare providers [45]. Men experience physical changes like fatigue, hair loss, incontinence, pain, hot flashes, and weight gain, which can significantly affect their quality of life [21,29,30,40]. Fatigue is a major factor in prostate cancer, and late middle-aged men struggle with normal life, leading to a decrease in their physical health [19,29]. Men with prostate cancer face intimacy and sexuality issues, fearing a loss of control [33]. Physical changes can lead to a loss of manhood and ego, and incontinence can cause loss and grief [19]. Public concerns about pads and washroom use because of incontinence are also prevalent, leading to decreases in social engagement and activities [30,33].

The psychological adjustment of men with prostate cancer is influenced by factors such as fear and anxiety, frustration, confusion, and depression, causing psychological imbalance [23,25,41,46,54]. Men often experience shock and anger at diagnosis, leading to distress and uncertainty, impacting their partners’ lives and their own [41,51]. Men experienced a normal state of mind after recognizing the impact of prostate cancer but often became obsessed with their PSA levels, leading to a cycle of relapsing–remitting anxiety [27,41]. Emotional support plays a role in men’s psychological well-being, with some preferring isolation and others acknowledging the importance of good social relationships to support mental state and promote quality of life [28,31,41,42,45]. Coping mechanisms include learning from multigenerational family histories, humor using traditional values, and engaging in proactive behaviors like preventive screening and healthcare services [20,23,25,32]. Emotional well-being affects engagement in medical consultations, and humor is a common male coping strategy [20].

While men and their partner experienced shock, trauma, disbelief, and fear upon hearing the diagnosis of cancer [38,49]. Partners adapt more fluidly to their roles, resulting in a more tenuous adjustment process [45,50,53]. Late-middle-aged couples face difficulties in adapting to prostate cancer, with physical, functional, and psychosocial concerns having the most negative effect on their quality of life [22,27,29,33,41]. It is evident that partners hide their emotions to protect their husband’s mental health and maintain their relationship [21,29,30,40].

#### 3.3.2. Relationships and Dynamics

Open communication is crucial in couples dealing with prostate cancer, as it helps manage the illness and promotes better understanding and support in their relationship [19]. Men often feel devalued due to a lack of funding, research focus, and awareness of prostate cancer [37,54]. Partners often initiate communication, as it is essential for managing the situation, expressing feelings, and sustaining relationship satisfaction [25,26]. Partners are challenged in supporting men with changing masculinity, with many feelings ill-prepared and uncertain [55]. Sexuality was difficult to discuss with healthcare professionals for men with prostate cancer due to poor interpersonal relation and communication [24]. Poor communication negatively impacts prostate cancer patients, causing false hope and confusion [24,25,54]. Partners are aware of the impact of prostate cancer on a man’s identity and masculinity but feel unprepared to manage it, thereby causing emotional distancing and shutdown [32,48]. Spirituality and partner support are effective coping mechanisms [44], and the impact on the family unit is also significant, with patients and partners differing on sexual-related matters [28]. Psychosocial interventions are needed to facilitate healthy communication and address sexual rehabilitation needs to maintain relationship stability [33]. Family support positively influences the experience with prostate cancer, and men’s perceptions of partners significantly impact their distress [29,32].

Prostate cancer treatment can lead to changes in sexual relationships, sensitive intimacy, and emotional familiarity among men [26,40,50,52]. Men and their partners may experience altered sexual patterns, while some insist on a sex life after treatment [29]. Stress can cause intimacy changes, leading to distance between couples [20,25,41]. Erectile dysfunction and lack of libido can cause impotence in late-middle-aged couples and sexual changes in young–old age couples, causing sexual and relationship dissatisfaction [26,53]. It is crucial to discuss intimacy and sex with oncology outpatient clinicians, despite limited opportunities and a lack of appreciation for hormone therapy [53]. The partner’s role in providing emotional and practical support to men increases due to prostate cancer and leads to greater cooperation and a healthier relationship [26].

#### 3.3.3. Treatment Journey

There is uncertainty regarding prostate cancer treatment options and outcomes, affecting mental well-being and quality of life [35]. The importance of listening with caution and being accountable for treatment decisions is emphasized to support patients and their partner in the treatment journey [45,52]. Some men report side effects such as dysuria, nocturia, and diarrhea, which make them uncomfortable during treatment [32,38,44]. The fear and anxiety men face when undergoing radical prostatectomies, leading to guilt, resignation, or exclusion from the gay community, is evident [51]. Additionally, half of men report negative effects on masculinity, sexual dysfunction, and erectile dysfunction from prostate cancer treatment [51]. Partners play a crucial role in providing emotional and practical support to men with prostate cancer during their treatment journey [26], and healthcare providers play a significant role in improving psychosexual adjustment and quality of life through psychosocial interventions [28]. Support groups and community networks are also essential for emotional and informational support [46,49], and some partners reported receiving more support from urologists [21]. Furthermore, prostate cancer education boosts patient confidence, group support, and healthcare provider reassurance, highlighting the importance of listening with caution and being accountable for treatment decisions [46].

#### 3.3.4. Survivorship and Aftercare

After prostate cancer, patients often face several challenges, including fear of recurrence, psychological adjustment, and lifestyle changes [51], with men often experiencing incontinence and erectile dysfunction [49]. While men in recovery require minimal physical care, this is provided by their partners [39], and partners often seek reassurance while providing this support during physical care after treatment [39]. Both men and their partners express concern about the effectiveness of treatment, where men suppress these feelings [27], and their partners have difficulty raising issues, so partners also suffer in silence, leading to loneliness and fear [18,51]. Some men experience altered sexual patterns due to the diagnosis and treatment of the condition [26,29]. Prolonged sexual incompetence also becomes a concern for participants after their treatment [27].

The partner often faces subjective distress, often more than their cancer partner, and regularly conceal their feelings and assure normalization [43,47]. Understanding sexual priorities and distress is crucial for better care and representation of the emotional journey [46]. Lifestyle changes include changes in self-image, loss of manliness, reduced self-esteem, and difficulty in new relationships [20]. Men after prostate cancer struggle with disclosing new circumstances and planning, and they cope with anxiety, stress, and worry by seeking a cancer-free life, engaging in activities, and spending time alone for mental clarity [18,35]. Feelings over sex loss and intimacy are common [41,46], with the partner often feeling excluded and ignored due to their partners’ focus on treatment and aftercare [29]. Understanding these challenges is essential for better care and representation of the emotional journey [34].

Post surgery care for patients with prostate cancer is often undervalued [43], with men often relying on humor and their partners as key support [20]. Men with prostate cancer often prefer telephone counselling [47], with partners often feeling relegated to the sidelines by healthcare providers, family, and friends [23]. Long-term monitoring is essential but could be improved by extending support to the pre-surgery phase and tailoring information to individual needs [48]. Postoperative urinary incontinence generally improves over time, becoming less of a problem than sexual dysfunction [47]. Psychosocial interventions can improve well-being and recovery and reduce the impact on relationships [54].

## 4. Discussion

This QES review aimed to synthesize the literature on the experiences and perspectives of persons with prostate cancer and their partner. This review found that prostate cancer treatment can be a double-edged sword. Whiles treating the condition, persons living with prostate cancer and their partners experience physical signs and symptoms like pain and fatigue, along with emotional or psychological distress like fear and anxiety. Partners become a crucial source of support; however, communication issues and a lack of preparedness for these challenges can cause strain in relationships.

The focus on survival rates often overshadows the profound physical toll that treatments exact on patients, impacting their daily lives and overall well-being. The physical impact and pervasive nature of fatigue among prostate cancer patients undergoing treatment cannot be underestimated [19,45]. Fatigue extends beyond mere tiredness, significantly disrupting a person’s ability to engage in routine activities [56]. Late-middle-aged men find themselves struggling to maintain their accustomed lifestyle due to this persistent exhaustion [55,57]. Incontinence has a distressing impact on patients [21,42]; beyond the physical discomfort, incontinence becomes a poignant symbol of loss and grief, often leading to a profound sense of emasculation and eroding self-esteem [58,59]. The fear of losing sexual function and intimacy is a pervasive concern among men with prostate cancer [29,33]. This fear extends to a loss of control over one’s masculinity, often causing emotional turmoil and strain within relationships [60]. The partners of patients with prostate cancer play a crucial role, yet they often conceal their emotional turmoil to protect their partners [40,42]. This emotional burden adds complexity to the overall impact on the patient’s and partner’s quality of life, contributing to a lack of comprehensive support [60,61]. Public concerns about using pads and frequent restroom visits due to incontinence persist, adding a layer of societal stigma and practical challenges [30,33], which exacerbate the already substantial psychological impacts on patients.

In terms of psychological well-being, a prostate cancer diagnosis significantly impacts the psychological well-being of men, often initiating a cascade of emotions including fear, anxiety, shock, and anger [23,25,32,41]. The recognition of this diagnosis often leads to a state of distress and uncertainty, affecting not only the individuals diagnosed but also their partners [62,63]. The cyclical nature of anxiety in men, particularly revolving around monitoring PSA levels, resulting in relapsing–remitting anxiety patterns cannot be underestimated [64,65]. To navigate this psychological turmoil, men often employ coping mechanisms rooted in familial values, humor, and proactive health behaviors [20,25,32,41,46]. The influence of emotional well-being on men’s engagement in medical consultations, emphasizing humor as a prominent coping strategy [66]. Furthermore, insights from recent studies [67] highlighted the significance of multigenerational family histories in learning adaptive coping mechanisms.

A clear finding of this review is the need for communication and support [19,24,25,54]. Insufficient funding and research on prostate cancer have led to men feeling undervalued, which is exacerbated by the psychological impact of the illness [46,48]. Open communication is crucial for managing the disease and fostering mutual understanding and support among partners [19]. However, poor communication, particularly regarding sensitive topics like sexuality, can lead to confusion and false hope among patients [68]. Partners often face challenges in supporting men dealing with changing masculinity and sexuality [69], which can be compounded by difficulties in discussing sexuality with healthcare professionals. Discussions about sexual relationships, which include spiritual support and partner support, are an effective coping mechanism for the illness [27], and tailored interventions targeting enhanced communication strategies and addressing masculinity changes and sexual rehabilitation needs are essential, yet people often feel unprepared to manage these changes [32,48]. The disease significantly affects the family unit, leading to differences in perceptions regarding sexual-related measures between patients and partners [28].

Navigating intimacy and sexuality after prostate cancer treatment is a complex journey that significantly impacts both men and their partners. The shifts in sexual dynamics, emotional closeness, and overall intimacy often become focal points after treatment. These changes can manifest differently for people and affect relationships [26,40,46,52]. For many partners, alterations in sexual patterns are a reality after treatment [26,53]. The struggle with erectile dysfunction, decreased libido, and the potential for impotence can disrupt the intimate bond that partners share. These changes can lead to varying responses, with some individuals and couples striving to maintain a fulfilling sex life, while others may grapple with distancing due to the stress and challenges [53]. Acknowledging these changes within the context of oncology care remains pivotal, yet it is often an overlooked aspect of treatment [70]. Limited opportunities and inadequate attention to discussing intimacy and sexual concerns, especially regarding hormone therapy, pose significant hurdles. Addressing these concerns within clinical settings could provide much-needed support and guidance for individuals navigating the period after prostate cancer treatment [71]. Moreover, the evolving roles of the partner in providing emotional and practical support to men affected by prostate cancer highlight a societal shift [72]. Partners often step into a crucial role, offering support and empathy as their partners navigate the challenges of treatment and post-treatment life [70].

Furthermore, the process of making medical decisions regarding prostate cancer treatment is fraught with uncertainty, impacting mental well-being and quality of life. The significance of careful listening and accountability in treatment decisions is essential [35,45,52]. This is crucial given the reported side effects of dysuria, nocturia, and diarrhea experienced by patients undergoing treatment, which significantly affect their comfort and daily lives [73]. The emotional toll of radical prostatectomies is evident, with men facing fears, anxiety, and subsequent feelings of guilt, resignation, or even exclusion from their communities [74,75]. The profound impacts on masculinity, sexual function, and the prevalence of erectile dysfunction after treatment showcase the multifaceted challenges individuals face [76]. However, amidst these challenges, the support system surrounding individuals dealing with prostate cancer emerges as a crucial factor [77]. Partners serve as pillars of emotional and practical support [26,46]. Healthcare providers also play a pivotal role by facilitating psychosocial interventions, enhancing psychosexual adjustment, and improving overall quality of life [78]. Support networks, including support groups and community connections, offer invaluable emotional and informational backing. Nonetheless, there. is a noted discrepancy where the partner may receive more support from urologists [21,28]. Addressing this sex-specific support bias becomes essential to ensure equitable support for both patients and their partners. Moreover, education about prostate cancer empowers patients, fosters confidence, encourages group support, and strengthens the patient–healthcare provider relationship [79].

Navigating life after prostate cancer treatment presents a myriad of challenges for both patients and their partners [30,44]. The fear of cancer recurrence looms large, impacting psychological adjustment and triggering significant lifestyle changes [39]. This phase often results in incontinence and erectile dysfunction for men, while partners seek reassurance and physical connection, emphasizing the different support needs within the relationship [43]. Recovery for men often involves minimal physical care, while the partner takes on roles encompassing support, household tasks, and emotional management [43]. Both men and their partners harbor concerns about treatment effectiveness, yet men tend to suppress their feelings while the partner suffers silently, leading to feelings of isolation and fear [39,52]. Sexual changes after treatment, including altered patterns and prolonged sexual incompetence, emerge as significant challenges for individuals grappling with their new realities [79]. The partner, experiencing subjective distress, often conceals their feelings, seeking to normalize the situation [23,34,73]. These emotional complexities underline the importance of understanding sexual priorities and distress to improve care and support. Lifestyle changes after prostate cancer treatment significantly impact self-image, self-esteem, and the ability to navigate new relationships [71]. Disclosing these new circumstances, planning, and coping with anxiety and stress after treatment become challenges for men [47,52]. This phase often leads to a pursuit of a cancer-free life, engagement in activities, and a need for solitude to attain mental clarity. In the realm of healthcare navigation, post-surgery care for prostate cancer patients tends to be undervalued, with men often relying on humor and spousal support [20,43]. Comparison to breast cancer care is made and the preference for telephone counselling highlight unique needs [47]. However, partners feel sidelined by healthcare providers, family, and friends, indicating a gap in the support system [79].

Long-term monitoring proves essential but requires improvement by extending support to the pre-surgery phase and tailoring information to individual needs [47,48]. While urinary incontinence after surgery has improved over time, sexual dysfunction remains a prominent issue [80,81]. Psychosocial interventions emerge as valuable tools in enhancing well-being, aiding recovery, and reducing the impact on relationships [54]. Understanding the diverse array of challenges faced by persons with prostate cancer and their partners after prostate cancer treatment is essential for tailoring comprehensive care that addresses the multifaceted emotional, physical, and relational aspects of this journey [75,80]. This evidence synthesis can help clinicians and institutions better understand the experience of patients with prostate cancer and their partners. The findings from this QES mainly focus on the presentation of traditional men and partners. However, society and relationships have changed or moved beyond the traditional partner, so future research should be focused on the need to include all especially marginalized communities like the LGBTQ+ community, and minority ethnic groups as they stress the need for individualized care and treatment strategies. Increased knowledge of the impact of diagnosis and treatment on both individuals and their relationships may encourage providers and clinicians to use psychological and therapeutic therapies to support mental health.

## 5. Strengths and Limitations

Using qualitative evidence synthesis helped us to conduct an in-depth analysis of the experiences of persons living with prostate cancer and their partners. The insights gained can assist in driving improvements in patient-centered care, support services, and inclusive representation, ultimately enhancing the overall well-being and quality of life of those affected by this condition. However, we recognize that qualitative thematic analysis has the potential for researcher bias and that this review focused only on qualitative studies, so this study may lack generalizability. Furthermore, this study may lack transferability to other contexts. Many papers did not include partners or same-sex relationship partners and mainly focused on the presentation of traditional relationships, which may not be representative of the society and relationships of today.

## 6. Conclusions

Based on the finding of this review, prostate cancer dramatically affects the experiences of individuals with prostate cancer and their partners. Prostate cancer significantly impacts patients and their partners, necessitating open communication about emotional and physical challenges, especially sexuality and treatment side effects. Comprehensive support systems addressing psychological, physical, and relational needs are essential, including access to counselling, support groups, and educational resources. Partners, when actively involved in care, require adequate information and support for their own struggles. Tailored, clear information about the disease, treatment options, side effects, and management should be provided to both patients and their partners. Long-term monitoring and follow-up systems are necessary for continuous support and intervention. Training healthcare providers to recognize and address the comprehensive needs of both parties with a holistic and empathetic approach is crucial, alongside encouraging ongoing research.

## Figures and Tables

**Figure 1 healthcare-12-01490-f001:**
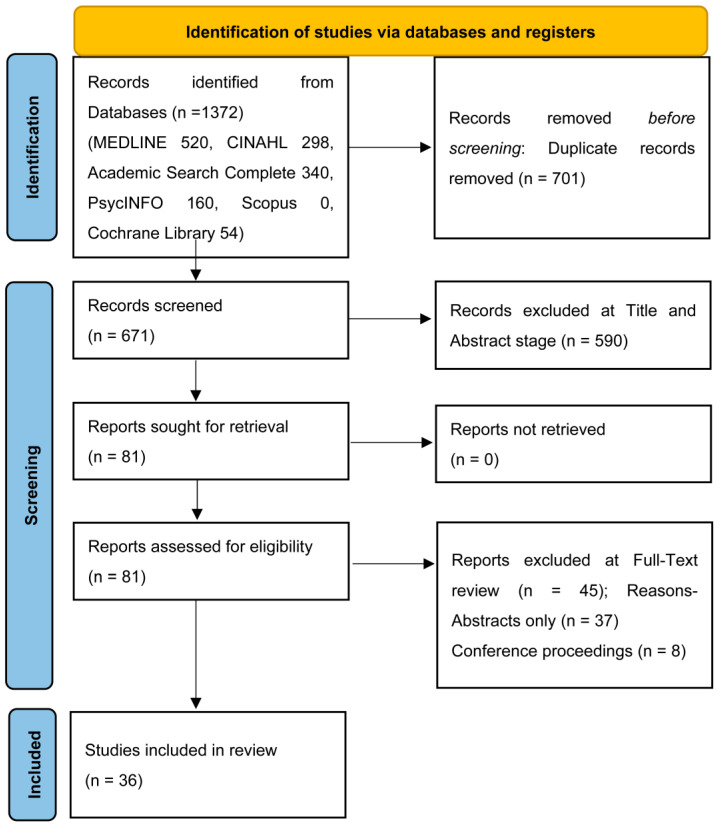
PRISMA 2020 flow diagram.

**Table 1 healthcare-12-01490-t001:** Search terms.

Search	Search Terms
S1	TI OR AB (MH “Prostate cancer”) OR TI OR AB (prostate cancer OR prostatic neoplasms OR prostate carcinoma)
S2	TI OR AB (partner* OR spous* OR husband* OR wif* OR wiv*OR significant other* OR coupl*)
S3	TI OR AB (experience* OR perception* OR attitude* OR view* OR feeling* OR opinion* OR emotion* OR perspective*)
S4	S1 AND S2
S5	S3 AND S4

**Table 2 healthcare-12-01490-t002:** Data analysis—codes, subthemes and themes.

Codes: Commonly Occurring Words or Phrases	Subtheme	Theme
Fatigue, hair loss, incontinence.	Physical impact	Quality of life
Fear, anxiety, stress, frustration, confusion, and depression.	Psychological well-being	
Open communication, lack of funding, ill-preparedness, uncertainty, poor communication.	Communication and support	Relationships and dynamics
Sensitive intimacy, emotional familiarity, erectile dysfunction, lack of libido, and impotence.	Intimacy and sexuality	
Uncertainty in treatment options, outcomes of treatment, and fear of decision making.	Medical decision making	Treatment journey
Emotional and practical support, healthcare provider roles, support groups, community networks, education, and reassurance.	Support system	
Fear of recurrence, psychological adjustment, lifestyle changes, incontinence and erectile dysfunctions, loneliness, fear, altered sexual patterns, long sexual incompetence, subjective distress, concealing feelings, and coping with anxiety.	Post-treatment challenges	Survivorship and aftercare
Post surgery care or support, postsurgical counselling, sidelines by healthcare providers, and long-term monitoring, improved by extending postoperative urinary incontinence over time and psychosocial interventions.	Healthcare navigation	

## Data Availability

The data in this review and the original published papers used in the review are the only available sources.

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
