# Peer review of "The Experiences and Perspectives of Persons with Prostate Cancer and Their Partners: A Qualitative Evidence Synthesis Using Meta-Ethnography"

_healthcare, 2024, doi:10.3390/healthcare12151490_

Round 1

Reviewer 1 Report

Comments and Suggestions for Authors

Thank you for allowing me to review your significant manuscript on prostate cancer, a topic of global importance. Your research is particularly intriguing, and I have some queries and observations that I believe could enhance your work. I trust that my feedback will be beneficial in refining your manuscript.

1.     In the abstract, in methods, you used “A QES” without the complete spell before this sentence. Authors usually do not use abbreviations in the abstract. Please use the fewest abbreviations in your abstract. Please reconsider and improve these abbreviations.

2.     Line 85, 2.5. Inclusion and exclusion criteria: I found no exclusion criteria in this part. If there are no exclusion criteria, you should explain why. Please reconsider and improve here.

3.     Table 2 and the results section show four themes in this research. I understand it is necessary to create abstract themes during the analysis process. However, it is hard to recognize that those themes relate to prostate cancer patients and their partners. If these results arise from the characteristics of this analysis method, you should add these research limitations to your manuscript. Because of these problems, the significance of this study's results is unclear. Please reconsider and improve these.

4.     Your manuscript and supplements contain many minor grammatical errors. Please recheck thoroughly and improve those.

Comments on the Quality of English Language

Including the above.

Author Response

Comment 1:  In the abstract, in methods, you used “A QES” without the complete spell before this sentence. Authors usually do not use abbreviations in the abstract. Please use the fewest abbreviations in your abstract. Please reconsider and improve these abbreviations.                                              Response 1: Thank you for your comments, we have explained the abbreviations to help provide greater clarity.

Comment 2: Line 85, 2.5. Inclusion and exclusion criteria: I found no exclusion criteria in this part. If there are no exclusion criteria, you should explain why. Please reconsider and improve here.      Response 2: We appreciate your recommendation and exclusion criteria has been added.

Comment 3: Table 2 and the results section show four themes in this research. I understand it is necessary to create abstract themes during the analysis process. However, it is hard to recognize that those themes relate to prostate cancer patients and their partners. If these results arise from the characteristics of this analysis method, you should add these research limitations to your manuscript. Because of these problems, the significance of this study's results is unclear. Please reconsider and improve these.                                                                                                                                      Response 3: We have acknowledged this as limitation of the review and added -A limitation of this review may be the use of qualitative thematic analysis which has the potential for researcher bias. As the process involves interpreting and categorizing data based on themes and the results can be influenced by the researcher's perspectives, preconceptions, and subjective judgments.

Comment 4: Your manuscript and supplements contain many minor grammatical errors. Please recheck thoroughly and improve those.

Response 4: Thank you, we have proofread the manuscript. 

Reviewer 2 Report

Comments and Suggestions for Authors

The authors conducted a qualitative evidence synthesis on the experiences and perspectives of persons with prostate cancer and their partners. Despite some parts seeming to be described accurately, I have several methodological concerns that undermine the reliability of the results. Some aspects are relevant and represent the basis for a scientific submission. The authors should also take better care of the presentation of the work, as some parts still appear to be in draft form.

Comments on the Quality of English Language

N.a.

Author Response

Comment 1: Abstract There is a typos error: “Abstract: Background:”.                                      Response 1: Thank you, we have proofread the manuscript and edited it.  

Comment 2: Introduction: Overall, the introduction contents are insufficient according to the writing rules for a scientific publication. Please organize the introduction in four to five paragraphs, with a logical flow from broad context to rationale formulation.                                                          Response 2: We have revised the introduction for flow and clarity.

Comment 3: The following statement is unnecessary: “The prostate gland is located in front of the rectum, below the bladder”. Please remove it and start the sentence with “Prostate cancer is the second most…” directly.                                                                                                               Response 3: We have removed the sentence as recommended.

Comment 4: Methods: The authors should present the entire search strategy used in the databases as supplementary material. Further, I suggest the authors remove “Table 1” from the text.           Response 4: Thank you for the suggestion, however, the entire search strategy has been presented in a supplementary document on figshare reference [14] and this is the link https://figshare.com/s/6fa4cdb704eac4de1d69 

Comment 5: Results; The authors cited Figure 1 in the text regarding the PRISMA flow diagram. However, this figure was not attached to the manuscript.                                                          Response 5: Figure 1 is now provided.

Comment 6: “31. Characteristics”: Please be more accurate in defining the subsection title. “Characteristics” is too general.                                                                                                  Response 6: 3.1 has been edited to Characteristics of included studies.

Comment 7: The authors should attach the results of the critical appraisal process. Table 2 should present graphically better. I suggest that the authors change the horizontal alignment of the text. In addition, the title of Table 2 cannot be “Table 2. Sub-themes and themes identified are presented in Table 2”.                                                                                                                                        Response 7: Table has been edited and Table label has been edited as recommended. 

Comment 8: The authors presented a paragraph with the characteristics of the studies in terms of design and methodological qualitative approach. However, the authors should present a table with the characteristics of the study in terms of authors, year, country, population characteristics, data collection modality and other relevant factors to better interpret the study’s findings. Part of this information must be included in a subsection in the text. The table presented as supplementary material does not have a title.                                                                                                     Response 8: The data extraction table addresses these concerns, and it is included as a supplementary file given its size and the fact that 47 studies were included in this study. 

Comment 9: Conclusions; The conclusion should be more concise                                            Response 9: Thank you for your feedback.

Reviewer 3 Report

Comments and Suggestions for Authors

This is an interesting and useful review.

It needs some additional information provided - the PRISMA chart showing how the inclusion/exclusion  criteria were applied.

It would also be useful to have the data and the findings separated by whether the focus is on the patient or the partner.

Author Response

Comment 1: It needs some additional information provided - the PRISMA chart showing how the inclusion/exclusion criteria were applied.                                                                                        Response 1: Thank you and the PRISMA has been included in the study.

Comment 2: It would also be useful to have the data and the findings separated by whether the focus is on the patient or the partner.                                                                                                      Response 2: This review focuses on both the person with prostate cancer and partners, and we have identified whether the finding refers to the person, partner or both. 

Reviewer 4 Report

Comments and Suggestions for Authors

Dear authors,

Many thanks for submitting your work to the journal. In my opinion, is a well-written, structured, and informative work that effectively addresses the experiences and perspectives of individuals' with prostate cancer using meta-ethnographic methods. However, I have made some comments on your work that must be addressed. Please see the attached file.

Best regards

Author Response

Comment 1: Although the discussion is quite informative and speculates the study's finding need to be organized better. First, give at the beginning a synopsis paragraph that summarizes the main findings of your study. Afterward, discuss the main findings by theme (Quality of life, Relationships and dynamics, Treatment journey, Survivorship / aftercare), following a logical order.              Response 1: A general synopsis of the review has been written before the discussion as suggested. Thank you for the recommendation.

Comment 2: Abstract: Please give the full form before the abbreviation QES (line 13).            Response 2: Thank you for correction the abbreviation has been written in full as recommended

Comment 3: Please provide the figure 1.                                                                                      Response 3: Figure 1 is provided. 

Comment 4: Please apply sub-sections to organize the section 3.3 Themes. Specifically, 3.3.1 Quality of life, 3.3.2 Relationships and dynamics, 3.3.3 Treatment journey, and 3.3.4 Survivorship and aftercare. Response 4: The themes are numbered as suggested, thank you. 

Round 2

Reviewer 2 Report

Comments and Suggestions for Authors

The authors did not fully address my concerns. Please check below for my previously mentioned concerns:

- the introduction remains insufficient in terms of contents and logical flow    (the authors reported only one paragraph)

- the conclusion is still too long

- the CASP tool was not provided or neither reported in the text

- The authors included a generic Supplementary File X in non-published materials. This file should be properly named and provided as supplemental material for publication.

Author Response

Comment 1: The introduction remains insufficient in terms of contents and logical flow (the authors reported only one paragraph).                                                                                                         Response 1: We have revised the introduction for flow and clarity.

Comment 2: The conclusion is still too long.                                                                                              Response 2:  Thank you for the suggestion and the conclusion has been revised to be more concise.

Comment 3: The conclusion is still too long.                                                                                              Response 3:  Thank you, the CASP critical appraisal range of scores are reported and is available on the open platform figshare [14].

Comment 4: The authors included a generic Supplementary File X in non-published materials. This file should be properly named and provided as supplemental material for publication.                                Response 4:  Thank you for your feedback, the file has been renamed and included as a supplementary file as suggested.

Reviewer 3 Report

Comments and Suggestions for Authors

The revised version now includes a PRISMA chart but the chart lacks detail. IT would be useful to include examples of the sort of papers which were eliminated at the abstract screening process.

The identification of the use of Thematic Analysis as a limitation suggests a ack of understanding about the analytic process.

Comments on the Quality of English Language

There are a few minor language issues.

Author Response

Comment 1: The revised version now includes a PRISMA chart but the chart lacks detail. IT would be useful to include examples of the sort of papers which were eliminated at the abstract screening process.                                                                                                                                            Response 1: Thank you and the PRISMA has been included in the study.

Comment 2:  The identification of the use of Thematic Analysis as a limitation suggests lack of understanding about the analytic process.                                                                                                 Response 2: Thank you for your feedback we have revised the information.
